# Wernicke’s Encephalopathy in Alcohol Use Disorder Patients after Liver Transplantation: A Case Series and Review of Literature

**DOI:** 10.3390/jcm9123809

**Published:** 2020-11-25

**Authors:** Gabriele A. Vassallo, Antonio Mirijello, Tommaso Dionisi, Claudia Tarli, Giuseppe Augello, Antonio Gasbarrini, Giovanni Addolorato

**Affiliations:** 1Department of Internal Medicine, Barone Lombardo Hospital, 92024 Canicattì, Italy; gabriele.vassallo86@libero.it (G.A.V.); dr.augello@alice.it (G.A.); 2Department of Medical Sciences, IRCCS Casa Sollievo della Sofferenza Hospital, 71013 San Giovanni Rotondo, Italy; antonio.mirijello@gmail.com; 3Alcohol Use Disorder and Alcohol Related Disease Unit, Department of Internal Medicine and Gastroenterology, Fondazione Policlinico Universitario A. Gemelli IRCCS, 00168 Rome, Italy; dionisitommaso@gmail.com (T.D.); claudia.tarli@gmail.com (C.T.); 4Internal Medicine, Gastroenterology and Hepatology Unit, Fondazione Policlinico Universitario A. Gemelli IRCCS, 00168 Rome, Italy; antonio.gasbarrini@unicatt.it; 5Institute of Internal Medicine and Gastroenterology, Catholic University of Rome, 00168 Rome, Italy; 6Internal Medicine Unit, Columbus-Gemelli Hospital, Department of Internal Medicine and Gastroenterology, Fondazione Policlinico Universitario A. Gemelli IRCCS, 00168 Rome, Italy

**Keywords:** Wernicke’s encephalopathy, Korsakoff syndrome, liver transplantation, alcohol use disorders, alcohol-related liver disease, alcohol-related cirrhosis

## Abstract

Wernicke’s encephalopathy (WE) is an acute neurological disorder resulting from thiamine deficiency, commonly found in alcohol use disorder (AUD) patients. Liver transplantation (LT) could represent a risk factor for the onset of WE in AUD patients, due to the onset of chronic depletion of thiamine in this population and the high metabolic demand of surgery and the postoperative period. However, few data are available about the risk of the onset of WE in AUD patients after LT. Here we report three cases of AUD patients who developed WE with confusion and delirium after LT. Prompt parenteral administration of thiamine led to a rapid improvement of the clinical condition and a complete remission of neurological symptoms after 3–4 days. In addition, a search of the available English literature was conducted in order to perform a review of the possible association between the onset of WE and LT in AUD patients. A prophylactic treatment regimen based on the administration of thiamine could be suggested in AUD patients before and after LT. Further studies are needed to determine the optimal regimen of thiamine in the prevention of WE in this setting.

## 1. Introduction

Wernicke’s encephalopathy (WE) is an acute, neuropsychiatric syndrome that is characterized by nystagmus, ophthalmoplegia, mental-status changes and the unsteadiness of stance and gait. The disorder results from a deficiency of thiamine and its biologically active form (thiamine pyrophosphate) that is an essential coenzyme in several biochemical pathways of the brain [1]. Untreated WE usually evolves towards Korsakoff syndrome, which is characterized by memory loss and confabulation [2,3]. WE is common in patients with alcohol use disorders (AUDs) [4]. Thiamine deficiency in AUD patients results from a combination of inadequate dietary intake, reduced gastrointestinal absorption, decreased hepatic storage and impaired utilization [5]. WE also occurs in the setting of poor nutrition caused by malabsorption (gastrointestinal surgery, including gastric bypass), poor dietary intake (hyperemesis of pregnancy, anorexia nervosa, dieting or intravenous feeding without proper supplementation of thiamine) or increased loss of the water-soluble vitamin thiamine (hemodialysis or peritoneal dialysis) [6,7,8]. In some cases, low levels of magnesium, an essential cofactor of thiamine into its active diphosphate and triphosphate forms, have been associated with thiamine deficiency in WE [9]. The precipitation of WE can occur during periods of high metabolic demand (acute severe systemic illnesses, pregnancy) or after administration of intravenous glucose [10]. To avoid this complication, the administration of thiamine prior to, or along with, glucose infusion is indicated, particularly in AUD patients or in those who are malnourished [4,11].

Clinical diagnosis of WE is frequently missed [5] because usually it coexists with other disorders that cause confusion, such as hepatic encephalopathy, sepsis and alcohol withdrawal syndrome [12,13]. A low blood thiamine level, lactic acidosis, elevated blood pyruvate or low transketolase activity can help to suspect thiamine deficiency. Thiamine blood levels may not reflect tissue stores and normal thiamine levels do not rule out WE [1]. Magnetic resonance imaging is currently considered the most valuable method to confirm a diagnosis of WE. Magnetic resonance imaging of the brain shows T2 and FLAIR hyperintensities in thalami, mammillary bodies, tectal plate and periaqueductal area [14]. Cerebrospinal fluid analysis is normal in most patients, although raised protein concentrations can occur in late stages. Electroencephalographic findings are within normal limits at an early stage but show no specific slowing of the dominant rhythm in a late stage [13]. Although laboratory tests and neuroimaging studies can be of help, the diagnosis of WE is primarily clinical. WE is diagnosed in patients that fulfill at least two of the following Caine criteria: (A) dietary deficiency; (B) oculomotor abnormalities; (C) cerebellar dysfunction; (D) altered mental status or mild memory impairment [15]. Although these criteria are more sensible for the diagnosis of WE, the classical triad of symptoms (ophthalmoplegia, ataxia and confusion) are more specific. Its diagnosis is also confirmed by the positive response of neurological signs to thiamine supplementation.

High doses of parenteral thiamine (200–500 mg t.i.d.) should be promptly administered in case of WE diagnosis or clinical suspicion to avoid progression toward Korsakoff syndrome or death [5]. Thiamine should be administered for 1–2 weeks, either intramuscularly or intravenously, because its gastrointestinal absorption in AUD or malnourished patients could be impaired [16]. The t.i.d. dosage regimen is based on the short half-life of thiamine [5]. Although anaphylaxis of intravenous administration is a rare event [16], to reduce this risk, thiamine can be administrated in 100 mL of normal saline over half an hour [17]. Multivitamin supplementation is also required in these patients [4]. Maintaining fluid and electrolyte balance is crucial in the management of these patients. Blood magnesium levels play an important role, since low levels have been implicated in the onset of WE and failure to respond to thiamine replacement [9].

Liver transplantation (LT) in AUD patients could represent a risk factor for the onset of WE, due to chronic thiamine depletion occurring in this population and high metabolic demand of surgery and postoperative period [18,19]. However, few data are available about the risk of the onset of WE in AUD patients after LT.

Here we report three cases of AUD patients that developed confusion and delirium 1–3 weeks after LT for end-stage alcohol-related liver disease. Based on their alcohol abuse history, malnutrition before transplantation, radiological findings, suggestive symptoms and prompt improvement after thiamine replacement, a WE diagnosis was made. In addition, a search of the available English literature (PubMed) was conducted in order to perform a review of the possible association between the onset of WE and liver transplantation in AUD patients.

## 2. Case 1

A 49-year-old man with a history of AUD developed altered mental status, with fluctuating global confusion, disorientation and agitation three weeks after LT was carried out for end-stage alcohol-related liver disease. The patient reported heavy alcohol consumption for several years (about 16 drinks/day). Total alcohol abstinence was achieved one year before the transplantation and no relapse or slip occurred in this period. His medication included proton pump inhibitors (PPI), propranolol, ursodeoxycholic acid, furosemide, methylprednisolone, tacrolimus and everolimus. Vital signs were normal. Physical examinations did not show any neurological focal deficits, and no fever was detected. No neck stiffness, miosis and/or mydriasis were found. Liver function was abnormal (alanine aminotransferease160 UI/L; total bilirubin 1,9 mg/dL; alkaline phosphatase 248 UI/L; γ-glutamyl transferase 131 UI/L). Renal function was unremarkable. Glucose and ammonium levels were normal. Electrolytes were normal except for hypokalemia (3.4 mEq/L; normal range 3.5–5 mEq/L) and hypomagnesemia (1.4 mg/dL; normal range 1.8–2.4 mg/dL). Cortisol and thyroid hormone levels were normal. Blood gas analysis did not show hypercapnia. Blood and urine cultures were sterile. Blood alcohol concentration was normal. Urinary benzodiazepine/opioid metabolites were negative. Levels of tacrolimus and everolimus were 3 ng/mL (normal range 5–15 ng/mL) and 3.5 ng/mL (normal range 5–15 ng/mL), respectively. Electrocardiogram showed a regular rhythm. Chest X-ray and abdominal ultrasound were unremarkable. An electroencephalogram showed mild generalized slowing rhythm. Brain magnetic resonance Imaging (MRI) on T2-weighted image showed hyperdensity of pale nucleus. The patient was treated with thiamine (200 mg t.i.d.) and haloperidol (5 mg/day) and showed improved clinical conditions and remission of neurological symptoms after three days.

## 3. Case 2

A 57-year-old man with a history of AUD and who was transplanted for alcohol-related cirrhosis developed delirium about two weeks after LT. His medication included PPI, furosemide, spironolactone, propranolol, ursodeoxycholic acid, tacrolimus, everolimus and mycophenolate. Physical examinations did not show any neurological focal deficits. There was no evidence of neck stiffness, miosis and/or mydriasis. Glucose and ammonium levels were normal. Electrolyte levels were normal. Liver function was normal. Renal function was unremarkable. Blood and urine samples did not show markers for possible infections. Blood alcohol concentration was absent. Urinary benzodiazepine/opioid metabolites were negative. Levels of tacrolimus and everolimus were in the normal range. Electrocardiogram showed a sinus rhythm. Chest X-ray was unremarkable. An electroencephalogram showed mild generalized slowing rhythm. Cranial MRI on T1-weighted image showed contrast enhancement of the medial thalami and around the periventricular region of the third ventricle. Clinical conditions improved and delirium disappeared after treatment with thiamine (300 mg/day). A complete remission of the altered mental status was detected after four days of thiamine administration.

## 4. Case 3

A 51-year-old man developed confusion, tremor and an altered gait with broad base one week after LT for end-stage liver disease related to alcohol abuse and hepatitis B infection. Notably, in the days following LT, he developed *Clostridium difficile* colitis and was treated with metronidazole and oral vancomycin. His medications at that time included PPI, propranolol, ursodeoxycholic acid, furosemide, methylprednisolone, tacrolimus and everolimus. Vital signs were normal and no fever was detected. Physical examinations did not show any neurological focal deficits. No evidence of neck stiffness, miosis and/or mydriasis. Liver function was markedly altered (ALT 212 UI/L; total bilirubin 14 mg/dL; alkaline phosphatase 84 UI/L). Renal function was unremarkable (creatinine 0.76 mg/dL). Glucose was 158 mg/dL and the ammonium level was normal. Electrolytes were normal except for hypocalcemia (8.2 mEq/L; normal range 8.6–10.2 mEq/L) and hypomagnesemia (1.6 mg/dL; normal range 1.8–2.4 mg/dL). Blood gas analysis did not show abnormalities. Blood and urine cultures were sterile. The level of tacrolimus was in the normal range. A chest X-ray and a crain computed tomography-scan were unremarkable. An electroencephalogram showed nonspecific abnormalities in the temporal lobes, and there were no signs that suggested an epileptic activity. Cranial MRI showed alteration of the hyperintense signal in the T2-dependent sequences of the transverse pontine fibers, with a “cross” appearance. However, no associated signs of atrophy of the pons and the bulb were observed. Atrophy was found in the cerebellum, with enlargement of the pericerebellar liquor spaces and the bulbar cistern. In the T1-dependent images, a blurred and symmetrical hyperintensity of the signal of the pale nuclei was observed. The patient was treated with thiamine (200 mg t.i.d.) and haloperidol (5 mg/day) and showed an improvement of clinical conditions and remission of neurological symptoms after three days.

## 5. Discussion

WE commonly develops in AUD patients as a result of thiamine deficiency, although it remains largely underdiagnosed. Thiamine deficiency in AUD patients is mainly due to inadequate dietary intake of the vitamin and diminished gastrointestinal absorption. However, liver disease may exacerbate thiamine deficiency, decreasing hepatic storage and impairing its utilization [20]. The 30% of patients with end-stage alcohol-related liver disease, who died because of neurological complications, presented a picture of WE at histological examination of the brain [21].

Some case reports and case series suggest a possible relationship between LT and WE [22,23,24]. LT in AUD patients could represent an additional risk factor for WE, since surgery and postoperative period represent a high metabolic rate condition, increasing the demand for thiamine [24]. In addition, the onset of surgery related to gastrointestinal disturbance (vomit, nausea and diarrhea) and total parenteral nutrition with high-carbohydrate diets after transplantation might contribute to the development of WE by reducing gastrointestinal absorption of thiamine and increasing its requirements, respectively. Okumura et al. reported case series of WE following excessive soft drink consumption due to poor thiamine concentration in these beverages and their high content of sugar which increases thiamine requirements [25].

Moreover, considering that thiamine body stores in healthy subjects are approximately 30 mg and that the estimated average requirement (EAR) for thiamine is 1 mg/day for men and 0.9 for women in a healthy condition, a state of depletion can develop within 18–20 days in patients with a high metabolic rate receiving a strict thiamine-free diet [26]. In the reported cases, patients developed neurological symptoms about 2–3 weeks after LT.

Hypomagnesemia has also been implicated with thiamine deficiency in onset of WE [9]. Hypomagnesemia is common both in AUD patients and in patients with advanced liver disease [27]. Loop diuretics can also lead to hypomagnesemia. In the reported cases, two patients reported hypomagnesemia, while the serum magnesium level of the third patient was unknown. All these patients were also treated with loop diuretics before and after LT. We hypothesized that hypomagnesemia might contribute to the onset of WE in these patients.

Calcineurin inhibitor could also contribute to the genesis of WE in transplanted patients, due to both its neurotoxicity and the increased demand for thiamine secondary to detoxification processes, mitochondrial decoupling and oxidative stress [28]. All three patients were treated with tacrolimus and everolimus, supporting this hypothesis.

In our cases, the diagnosis of WE was supported by radiological findings. Typical MRI findings include areas of increased T2 and FLAIR signals, decreased T1 signal and diffusion abnormality surrounding the aqueduct and third ventricle and within the medial thalamus, dorsal medulla, tectal plate and mamillary bodies [29]. Lesions may also be seen in atypical areas such as the cerebellum, cranial nerve nuclei, dentate nuclei, caudate, red nuclei, splenium and cerebral cortex [29]. According to the largest series of patients with WE published, only 53.6% of patients, for whom magnetic resonance imaging data were available, had WE-related lesions [30].

After thiamine replacement, clinical conditions of patients improved, and there was remission of neurological symptoms. No other drugs were administered, except for haloperidol, which was administered to two patients to control agitation.

Similar to literature data showing that AUD patients presented cerebellar signs more frequently than ocular signs, only one patient (Case 3) in our case series presented tremor and ataxia, and none of the patients showed ophthalmoplegia [30].

A possible explanation for the onset of WE about two weeks after LT could be the presence of surgery related to gastrointestinal disturbance (vomit and diarrhea) and total parenteral nutrition after transplantation that contributed to the thiamine deficiency reducing its intake and increasing its requirements, respectively. This chronological relationship between the onset of WE and LT was also reported by Xye et al., who described a case of a male patient who developed WE two weeks after he received orthotopic liver transplantation because of hepatitis-B-related cirrhosis. Another case report described the onset of WE 27 days after allogeneic stem cell transplantation because total parenteral nutrition was administered after transplantation [31].

In summary, the pathophysiology of thiamine deficiency in patients transplanted for alcohol-related cirrhosis is characterized by multiple factors: (A) thiamine deficiency and malnutrition due to a long history of alcohol abuse; (B) decreased ability to store thiamine and activate its biologically active form due to liver cirrhosis; (C) insufficient thiamine intake due to thiamine-free diet (e.g., total parenteral nutrition), low food intake after surgery and gastrointestinal disturbance (vomit, nausea and diarrhea); (D) increased thiamine requirement due to high-rate metabolic conditions (LT and rehabilitating post-surgery); (E) hypomagnesemia induced from large use of loop diuretics; (F) use of calcineurin inhibitor. These cases reported highlight the role of liver cirrhosis and surgery in the development of WE aside from chronic alcohol abuse. In fact, all patients were abstinent at the time of liver transplantation, and before that, they were evaluated by alcohol specialists and were regularly tested with markers of alcohol abuse [32].

## 6. Conclusions

WE is a possible complication of LT in AUD patients. The evidence suggests that physicians should pay more attention to WE after LT to avoid delaying treatment. Moreover, prophylactic treatment with thiamine could be useful in all AUD patients before LT in order to prevent the onset of WE, given its safety profile, low cost and availability. A prophylactic treatment regimen based on available literature data could be the administration of thiamine (100–300 mg daily) before and after LT [5]. Further studies are needed to determine the optimal regimen of thiamine in the prevention of WE in this setting.

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
