# Peer review of "Wernicke’s Encephalopathy in Alcohol Use Disorder Patients after Liver Transplantation: A Case Series and Review of Literature"

_jcm, 2020, doi:10.3390/jcm9123809_

Round 1
Reviewer 1 Report
Thank you for the opportunity to review "Wernicke's Encephalopathy in Alcoholic Patients after Liver Transplantation: A Case Series and Review of the Literature." This paper includes a thorough description of Wernicke's from biological and clinical perspectives followed by three cases of liver transplant recipients who underwent OLT and were deemed to have WE.
---I would suggest that the authors move away from "alcoholic" as a descriptive term. The field is doing all it can to reduce stigma in language. "Alcohol-related" is a more neutral alternative. [Asrani SK, Trotter J, Lake J, Ahmed A, Bonagura A, Cameron A, DiMartini A, Gonzalez S, Im G, Martin P, Mathurin P. Meeting report: the Dallas consensus conference on liver transplantation for alcohol associated hepatitis. Liver Transplantation. 2020 Jan;26(1):127-40.]
---The authors will be well aware of the correlation between alcohol use disorder (AUD), alcohol-related liver disease (ALD), and gastric bypass procedures. I wondered if gastric bypass was included in the "gastrointestinal surgery" label in the first paragraph? If so, including it as an example (i.e. "gastrointestinal surgery including gastric bypass") might be helpful to readers since many of us treat patients who both overeat and drink excessively and WE is on our minds.
--One of the main questions I had reading the paper was the chronological relationship between drinking and WE. Case 1 had been abstinent for 1 year prior to transplant. We are not told how long Case 2 and 3 had been abstinent. This chronological relationship between drinking and WE is not treated in the introduction either. Under what conditions would a patient who had stopped drinking for prolonged periods of time still be vulnerable to developing WE? Had Case 2 and 3 stopped drinking? Any alcohol biomarkers (apart from serum ethanol) to verify this?
---Were there other nutritional deficiencies (i.e. poor diet or other malabsorption patterns) in these patients that could account for why thiamine stores might not have been stabilized or repleted between the time of alcohol cessation and the acute presentation peri-operatively? Were patients given multivitamins or other supplementation peri-operatively?
--Without giving some additional details as to why these presentations had to be WE, some of us who watch mental status changes happen regularly peri-operatively around liver transplant may just say that these presentations might have just been delirium (numerous contributors peri-operatively including anesthesia, immunosuppressants, pain, sleep irregularities, etc.) with some overlapping lab and imaging findings with WE.
The paper is compelling but there are a few gaps that need to be filled in, in my view. Since ALD is the number 1 indication for liver transplant in the USA and elsewhere, what are the authors proposing that we do? This is where some additional description of the chronological relationship between drinking and WE needs to be filled in. All ALD patients undergoing transplant need thiamine? Some? Which?
Thanks for the opportunity to review this important paper.
Author Response
Reply to reviewer 1
We have modified the term alcoholic with Alcohol Use Disorder patients or with Alcohol-related Liver Disease depending on the context.
We have added the following words “including gastric bypass” as example for malabsorption given the correlation between this procedure, eating disorders and alcohol use disorders.
All patients awaiting liver transplantation in our center were evaluated and followed by alcohol specialist that had monitored alcohol abstinence. Before liver transplantation all patients were regularly tested with markers of alcohol intake according current clinical guidelines. Regarding thiamine deficiency in alcohol use disorder patients after liver transplantation, its pathogenesis is multifactorial and due to also to malnutrition and liver cirrhosis, other than the high metabolic demand of surgery for liver transplantation.
Malabsorption due to alcohol abuse is common in alcohol use disorder patients. Moreover, the diet of these patients is usually poor of micronutrients and vitamin. Liver cirrhosis is also responsible for decreased hepatic storage of thiamine and its impaired utilization. The reported patients were no supplemented with thiamine before liver transplantation.
Delirium is a frequent condition occurring after liver transplantation. But in reported cases, the symptoms, radiological findings and the response to thiamine supplementation are suggestive of a diagnosis of Wernicke encephalopathy.
Additional descriptions of the chronological relationship between onset of Wernicke encephalopathy and its risk factors were given in discussion.
Given its safety profile, low cost and availability, thiamine should be administered to all Alcohol Use Patients awaiting liver transplantation.
Reviewer 2 Report
The authors present an interesting series of 3 liver transplant patients who developed symptoms and radiological alterations consistent with Wernicke encephalopathy. Although there are only 3 patients, the possibility of using prophylactic doses in these patients is postulated. Furthermore, the justification for the thiamine deficiency associated with anticalcineurinics is also interesting.
Major Comments:
1.-The title of the article speaks of “review of the literature”, but the previous published cases are only mentioned in one sentence and are not compared in the discussion with the 3 exposed (use of anti-calcinerin drugs, reversal without sequelae, delay in thiamine initiation, etc…). Although perhaps the best would have been a systematic review, I believe that the possibility of review of the literature can be accepted.
Minor comments:
1.- It should be mentioned in the introduction that to date, the classic triad is still present to classify a patient with Wernicke's encephalopathy. It is more specific but less sensitive than the Caine criteria.
2.- The reference of the largest series of patients with Wernicke encephalopathy published to date should be included, which compares the differences between alcoholic and non-alcoholic patients, and which in many aspects coincides with different paragraphs of the discussion and even of the introduction (https://doi.org/10.1016/j.mayocp.2017.02.019).
3.-Among the conclusions, it might be interesting to include that the determination of thiamine levels before and after liver transplantation in alcoholic patients can avoid complications. But most important is the possibility of Wernicke encephalopathy should be included in cases of neurological alterations in these patients.
Author Response
Reply to Reviewer 2
More details about other previous cases reported was given in discussion section.
We have specified the difference, in terms of specificity and sensitivity, between classical triad symptoms and Caine Criteria.
The study comparing alcoholic and non-alcoholic patients was added in the discussion paragraph.
We have added that physicians should pay more attention to the diagnosis of WE after LT in order to avoid delaying treatment, as suggested by reviewer. To the best of our knowledge, we don’t believe that determination of thiamine levels can help physicians because these levels don’t reflect the tissue store of thiamine.
Reviewer 3 Report
This is a well-written report on Wernicke Encephalopathy following liver transplantation in alcoholic patients.
I have a number of concerns that need to be challenged:
- In the introduction, the link between Korsakoff's syndrome and untreated Wernicke's Encephalopathy is explained. There are a number of issues regarding this topic:
1. The adequate treatment of Wernicke's Encephalopathy is 500mg IV/IM, three times per day, according to the guidelines by Thomson, Marhall, Bell, 2012 (Time to Act on the Inadequate Management of Wernicke's Encephalopathy in the UK). According to this guideline, the treatment in all three cases in the review is too low. I find this topic very important, because there is a dose-dependent outcome relation with treating with lower doses of thiamine and the development of chronic issues (e.g. Korsakoff's syndrome): see Oudman, Wijnia, Oey, van Dam, Painter, Postma, 2019: "Wernicke's encephalopathy in hyperemesis gravidarum: a systematic review". In this paper the group with higher doses (>500mg IV/IM) had fewer Korsakoff patients, and less fatalities.
2. Prophylaxis of Wernicke's Encephalopathy of at risk individuals is necassary. 100mg daily IV/IM in at risk individuals is suggested in the above mentioned papers. This would be good to apply in all liver transplant cases with alcoholic background. It is uncacceptable that patients suffer from malnutrition in a hospital setting, leading to acute and even chronic neurological disorders.
3. The support of WE diagnosis with neuroimaging is controversial, because not all patients have altered brain scans (specifically CT is not sensitive, but also MRI is not sensitive with conventional parameters and without a specific focus on thalamic areas).
4. In the discussion possible causes of WE are mentioned. the thiamine-free diet can also be a thiamine deficient diet. See for example the case series on wernicke following excessive soft drink consumption ( Okumura, Ida, Mori, Shimizu, ...2018) and the baby formula without thiamine (Prensky, 2005). In the soft-drink cases, high consumption of sugar required more than available thiamine.
Minor:
check for language. For example "hystory" , "die" "conditions (with underlined u)" / Urinary benzodiazapine/opioid metabolites were negative.
Author Response
Reply to Reviewer 3
As we reported in the introduction paragraph ideal dose of treatment for Wernicke Encephalopathy is 200-500 mg t.i.d, but unfortunately the reported patients have been undertreated
We totally agree that all liver transplant cases with alcoholic background should be supplemented with thiamine, as specified in the original version of the paper, both in the abstract and in the conclusion sections.
We agree that diagnosis of Wernicke Encephalopathy with neuroimaging is controversial and we have added data of large sample study showing as only 53.6 % of patients with this complication had typical radiological findings
We have discussed also about thiamine-free diet related to excessive consumption of soft drinks as suggested
We have checked English language.
Round 2
Reviewer 1 Report
Thank you for the opportunity to review a revised draft of "Wernicke's Encephalopathy in Alcohol Use Disorder Patients after Liver Transplantation: A Case Series and Review of the Literature."
The authors have strengthened the paper in terms of providing additional citations and background as to how WE and thiamine depletion can occur in non AUD patients (new citations # 25, 31) as well as AUD patients who are not actively drinking. They argue in their responses to author feedback that they feel more generic delirium was not the most likely cause of these patients' altered mental status.
There are still some scattered type-os and places where grammar could be polished and tightened.
This is an important issue for the field to consider as ALD emerges as a major, if not leading, indication for OLT worldwide.
This manuscript is a resubmission of an earlier submission. The following is a list of the peer review reports and author responses from that submission.